chemical engineering

mother liquor, gas field wastewater, electro-Fenton, initial pH, organic matter removal mechanism

**Author for correspondence:**
Hui-qiang Li
e-mail: lhq_scu@163.com

This article has been edited by the Royal Society of Chemistry, including the commissioning, peer review process and editorial aspects up to the point of acceptance.

# Organic matter removal from mother liquor of gas field wastewater by electro-Fenton process with the addition of $H_2O_2$: effect of initial pH

## Yan Wang, Hui-qiang Li and Li-ming Ren

College of Architecture and Environment, Sichuan University, Chengdu 610065, People's Republic of China

H-qL, 0000-0002-9008-3596

The electro-Fenton (EF) process was applied to treat mother liquor of gas field wastewater (ML-GFW). The Fe-Fe electrodes were used and $H_2O_2$ was added to the EF system. Effect of initial pH on chemical oxygen demand (COD) removal efficiency, specific electrical energy consumption (SEEC), specific electrode plate consumption (SEPC) and organic matter removal mechanism was investigated. The results showed that COD removal efficiency reached the maximum (71.9%) at initial pH of 3 after reaction for 3 h. Besides, considering with the SEEC and SEPC, pH of 3 was also the best choice, at which SEEC was 4.7 kW h $kg_{COD}^{-1}$, SEPC was 0.82 kgFe $kg_{COD}^{-1}$. Organic matter removal was achieved by two ways: oxidation and flocculation, and oxidation played a major role. With the analysis of GC-MS, the possible degradation pathways of the representative contaminants in the ML-GFW were given.

## 1. Introduction

In recent years, the consistently high growth rate of Chinese economy is supported by energy consumption which is heavily dependent on oil-based resources [1,2]. The petroleum and gas field mining industry is still blooming and plays a key role in global industrial development [3], and the generation of gas field wastewater must not be underestimated.

Gas field wastewater treatment technologies include physical methods, biological methods and physico-chemical methods mainly. For physical methods, membrane technologies are

expensive to apply [4], while thermal technologies [5] seem to be the suitable pretreatments for gas field wastewater to reduce the volume of the wastewater, and multistage flash, distillation, vapour compression distillation and multi-effect distillation are the major thermal technologies. The applications of the biological treatments on gas field wastewater are often challenged by poor load and impact resistance ability, and microbes are sensitive to the toxicity of pollutants in the wastewater [6]. Regarding physico-chemical methods, advanced oxidation processes (AOPs) are effective for the treatment of wastewater that contains non-biodegradable, inhibitory or toxic compounds [7] such as the gas field wastewater. AOPs can be classified into four categories: homogeneous chemical oxidation ($H_2O_2/Fe^{2+}$ and $H_2O_2/O_3$), photocatalytic processes ($H_2O_2/UV$ and $TiO_2/UV$), sonochemical oxidation and electro-chemical oxidation [8]. As one of the most efficient electro-chemical oxidation treatments, the application of electro-Fenton (EF) processes for gas field wastewater treatment is still limited.

As Yang & Xiang [6] have studied that cryogenic multi-effect distillation can be selected for desalination and deep treatment of gas field wastewater. After distillation, distilled condensate water quality can satisfy the grade one standard specified in the Integrated Wastewater Discharge Standard (GB 8978–1996), but the remaining gas field wastewater is concentrated and can be called mother liquor of gas field wastewater (ML-GFW). Raw gas field wastewater is characterized by high salinity and the presence of sulfides, ammonia nitrogen and complex organic matter, thus after distillation, ML-GFW seems difficult to treat [9]. Without effective treatment, ML-GFW will severely pollute estuaries, rivers, lakes, soil and even the air [10]. However, there have been few types of research on the treatment of ML-GFW; study and selection of suitable methods for ML-GFW treatment are both a challenge and an opportunity [11]. The ML-GFW contains high concentrations of refractory organic matter and $Cl^-$, which cannot be well treated by physical methods or biological treatment solely. In addition, the high conductivity of ML-GFW is really beneficial to the EF treatment, thus EF treatment would be a suitable process for ML-GFW treatment.

EF treatments can be divided into four types according to the addition or generation of Fenton's reagent [12,13]. In type 1, hydrogen peroxide and ferrous ion are both electro-generated. In type 2, hydroxyl radical is generated by Fenton reagent externally added, and ferrous ion is regenerated through the reduction of ferric ions on the cathode. In type 3, ferrous ion is externally added, and hydrogen peroxide is electro-generated. In type 4, hydrogen peroxide is externally added, while ferrous ion is electro-generated. Compared with the other three types, the last type is easier to control and can quickly get plenty of hydroxyl radicals. Since the chemical oxygen demand (COD) of ML-GFW was about $1.54 \times 10^4\,mg\,l^{-1}$ in this study and the hydroxyl radicals were in great demand, thus the last type was used. The electrode plates were made of iron in this study, which can constantly generate ferrous iron under power-on conditions (equation (1.1)), while $H_2O_2$ was added regularly into the system to ensure the treatment efficiency of ML-GFW. The main reactions in EF treatment are summarized as follows [14,15]:

Anode:

$$Fe \rightarrow Fe^{2+} + 2e^- \tag{1.1}$$

$$4OH^- \rightarrow O_2{\uparrow} + 2H_2O + 4e^- \tag{1.2}$$

Cathode:

$$2H_2O \rightarrow H_2{\uparrow} + 2OH^- \tag{1.3}$$

$$Fe^{3+} + e^- \rightarrow Fe^{2+} \tag{1.4}$$

In the wastewater:

$$H_2O_2 + Fe^{2+} \rightarrow Fe^{3+} + OH^- + \cdot OH \tag{1.5}$$

$$OH + organics \rightarrow products \tag{1.6}$$

$$2H_2O_2 \rightarrow 2H_2O + O_2{\uparrow} \tag{1.7}$$

$$Fe^{2+} + 2OH^- \rightarrow Fe(OH)_2\,(gel) \tag{1.8}$$

$$Fe^{3+} + 3OH^- \rightarrow Fe(OH)_3\,(gel) \tag{1.9}$$

The pH in the EF system plays a vital role because it can control the activity of oxidant and substrate, speciation factor of iron (equations (1.8) and (1.9)) and $H_2O_2$ (equation (1.5)) [13]. In addition, there is a lack of studies on ML-GFW treatment, let alone the effect of initial pH on treating ML-GFW by EF. In this study, ML-GFW was the target wastewater, and the effect of initial pH on COD removal performance by the EF process was mainly studied, and the organic matter removal mechanism was also analysed. Meanwhile, specific electrical energy consumptions (SEEC) [16] and specific electrical plate

consumptions (SEPC) were calculated for the assessment of economic aspects. Moreover, GC-MS was used to identify the intermediates of the organic matter in the ML-GFW during EF process, and the detailed degradation mechanism for the representative contaminants was proposed. Compared with previous studies, this paper provides a theoretical basis for the application of EF in actual ML-GFW.

# 2. Experimental set-up

## 2.1. Experimental materials

$H_2O_2$ (purity of 30%) was used as chemical oxidant purchased from Aladdin Ltd Co., Shanghai, China. The initial pH of the ML-GFW was adjusted by adding analytical grade sulfuric acid and sodium hydroxide purchased from Aladdin Ltd Co., Shanghai, China. All solutions were prepared with ultrapure water. Iron electrodes were one of the most common materials used in the EF system, which were used in this study. CV curves of the iron electrodes could be achieved in the study of Hu *et al.* [17]. The real ML-GFW obtained from southwest of China and its characteristics are summarized in table 1.

## 2.2. Electro-Fenton apparatus and procedures

The whole experimental device consisted of three parts: power supply, magnetic stirring device and reaction device. Diagram of EF apparatus is shown in figure 1. DXN-F AC power supply (Jiangyin Daheng Electric Co., Ltd) was used in the EF apparatus. Its output voltage ranged from 0 to 100 V, and output current ranged from 0 to 10 A. In this study, the ML-GFW was processed in a constant current of 2.0 A. The magnetic stirring device was used with a constant speed of 400 r.p.m. to ensure the reaction solution in a uniform concentration. As for the reaction device, it consisted of an electrolytic cell and two iron electrode plates. ML-GFW (500 ml) was adjusted to the target pH and added into the electrolytic cell. The iron electrode plates were ground with abrasive paper and immersed in 15% hydrochloric acid for 2 min and then rinsed with deionized water to remove the passivation layer. After that, the iron electrode plates were dried and weighed ($m_{Fe1}$). In addition, the iron electrode plates were also cleaned by deionized water, dried and weighed ($m_{Fe2}$) after the reaction. Two prepared iron electrode plates were placed in parallel, with a fixed plate spacing of 2.0 cm. The iron electrode plate specifications were $40 \times 60 \times 3$ mm (with a plate area of 24 cm$^2$).

The reaction time for each experiment was 3 h. In order to ensure that there were sufficient chemical oxidants in the reaction solution, 7 ml $H_2O_2$ was added every half an hour. Samples were taken from the electrolytic cell every half an hour. After filtration with 0.45 μm filter membrane, COD was examined and recorded as $COD_{(aq.)t}$. In the energized state, ferrous ions were produced at the anode and the flocs were formed and a part of organic matter was adsorbed by the flocs. The effect of flocculation on COD removal ($COD_{(fl.)t}$) was also analysed.

## 2.3. Analytical methods

The pH was monitored by a portable pH meter (Leici PHS-25, China). COD and consumption of iron electrode plate were measured according to standard methods. COD in flocs could be used to evaluate the contribution of flocculation to the removal of organics [18]. Thus, the organic matter removed by flocculation were measured as follow. After reaction for 3 h, 100 ml reaction solution was taken and centrifuged at 5000 r.p.m. for 10 min, and the supernatant was removed after centrifugation. Then, the remaining flocs were washed and centrifuged with 10% dilute sulfuric acid several times, and the supernatant was retained for COD measurement ($COD_{(fl.)t}$). The main calculations are listed below:

$$CODR_t(\%) = \frac{COD_{(aq.)0} - COD_{(aq.)t}}{COD_{(aq.)0}} \times 100\%. \tag{2.1}$$

$CODR_t$, total removal efficiency of COD after reaction for time $t$; $COD_{(aq.)0}$, COD concentration in solution before reaction; $COD_{(aq.)t}$, COD concentration in solution after reaction for $t$.

$$CODR_{(fl.)t}(\%) = \frac{COD_{(fl.)t}}{COD_{(aq.)0}} \times 100\%. \tag{2.2}$$

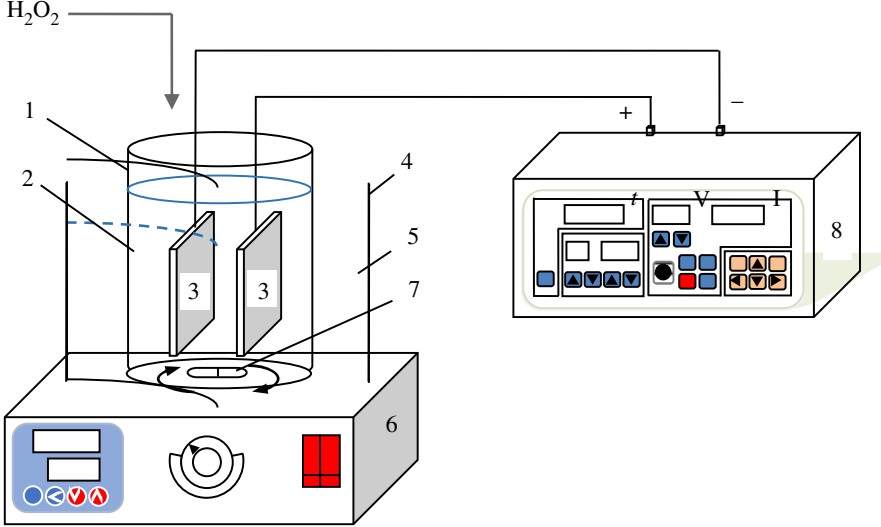

**Figure 1.** Diagram of EF apparatus. (1) Electrolytic cell, (2) ML-PFW, (3) iron electrode plates, (4) water bath, (5) distilled water, (6) magnetic stirrer, (7) magnetic bar, (8) cycle conversion power supply: V, volt meter; I, ampere meter; $t$, the time for periodical reversal of electrodes.

**Table 1.** Characteristics of the real ML-GFW.

| parameter | value | units |
|---|---|---|
| pH | 7.42 | |
| COD | $1.54 \times 10^4$ | mg l$^{-1}$ |
| oil | 9.36 | mg l$^{-1}$ |
| conductivity | $4.76 \times 10^5$ | μS cm$^{-1}$ |
| chloride ion | $1.40 \times 10^5$ | mg l$^{-1}$ |
| total iron | 3.02 | mg l$^{-1}$ |
| HCO$_3^-$ | $2.84 \times 10^3$ | mg l$^{-1}$ |

CODR$_{(fl.)t}$, removal efficiency of COD in flocculation after reaction for $t$; COD$_{(fl.)t}$, COD concentration in flocs after reaction for $t$.

$$CODR_{(oxi.)t}(\%) = CODR_t - CODR_{(fl.)t} \qquad (2.3)$$

CODR$_{(oxi.)t}$, removal efficiency of COD in oxidation after reaction for $t$.

$$SEEC(kWh\ kg^{-1}(COD)) = UIt/1000\Delta COD \times 0.5. \qquad (2.4)$$

SEEC, specific electrical energy consumption; U, voltage, V; I, Current, A; $t$, running time, h; $\Delta COD$, change in COD before and after treatment, mg l$^{-1}$; 0.5, volume of wastewater, l.

$$SEPC\left(\frac{kg_{Fe}}{kg_{COD}}\right) = \frac{m_{Fe1} - m_{Fe2}}{\Delta COD}. \qquad (2.5)$$

SEPC, specific electrode plate consumption; m$_{Fe1}$, mass of Fe electrode before treatment; m$_{Fe2}$, mass of Fe electrode after treatment; $\Delta COD$, mass of removal of COD, kg.

The samples were taken out of the electrolytic cell after reaction at different pH values and extracted by organic solvents for GC-MS analysis. The samples obtained from different pH values were 60 ml each and were divided into three parts (20 ml each). The pH values of the three parts were adjusted to be 2, 7 and 12, respectively. After that, 10 ml of dichloromethane was added to the separation funnel and shaken for 5 min. The organic liquid in the lower layer was collected and repeated three times. In the end, the extracted liquid was merged; 7890/5975 GC-MS system (Agilent, USA) was used to identify the intermediates.

# 3. Results and discussion

## 3.1. COD removal performance

The pH in the EF process can control the activity of oxidant and substrate, speciation factor of iron and $H_2O_2$ so that it can be considered as a pH-dependent process [13]. In figure 2a, during the electrolysis time, pH changed a little. The amount of $HCO_3^-$ in the ML-GFW was $2.84 \times 10^3 \, mg \, l^{-1}$ in table 1, and the buffering capacity of $HCO_3^-$ was of great influence to the experiment, because it contributed to the stabilization of $H_2O_2$ and resulted in prolonged reactivity [19].

COD removal efficiency was calculated to describe organic matter removal performance in the EF process. Figure 2b shows the apparent COD removal efficiency in the EF system with initial pH varied from 2 to 8 in treating the ML-GFW. Initial pH value had a vital influence on COD removal performance of the EF system. COD removal efficiency was 47.4%, 71.9%, 71.8%, 66.7%, 61.8%, 59.9% and 58.7% at reaction time of 3 h when pH value was 2, 3, 4, 5, 6, 7 and 8, respectively. Fernandes et al. [20] reported that when the sanitary landfill leachate was treated by EF oxidation, the maximum COD removal efficiency was 40%. The EF system applied in this study was more efficient on COD removal. To accurately characterize the COD removal efficiency, the influence of initial pH on the kinetics of the apparent COD removal was carried out The results described by the first-order equation are shown in figure 2c [21]. High values of the $R^2$ (higher than 0.97) demonstrated that the first-order model was satisfactory to represent the COD removal efficiency of ML-GFW by the EF process. According to the value of $K_{obs}$ ($min^{-1}$) in figure 2d, the fastest efficiency was achieved at pH of 3, and it sharply decreased to the lowest efficiency at pH of 2. The result can be explained by Ma et al. [22] that the Fenton reaction had the highest catalytic activity at pH = 2.8–3.0, and exceptionally low pH would inhibit the reaction between $Fe^{3+}$ and $H_2O_2$. In addition, the pH lower than the optimum values would induce the scavenging effect of $H^+$ on $OH\cdot$ [23], and made the activity of $[Fe(H_2O)_6^{2+}]$ higher than $Fe^{2+}$. The reaction between $[Fe(H_2O)_6^{2+}]$ and hydrogen peroxide was slower compared with $Fe^{2+}$ and thus generated a lower amount of hydroxyl radicals [23,24], which induced the COD removal efficiency by oxidation to be weakened. In addition, when pH was lower than 3, $Fe^{3+}$ could not be precipitated as $Fe(OH)_3$, so the flocculation was also weakened [25]. Except pH of 2, COD removal efficiency decreased with the increase of pH value. It was due to the formation of $Fe^{2+}$ and $Fe^{3+}$ complexes at higher pH values which led to a drop of $Fe^{2+}$ concentration (equation (1.8)) [26]. $H_2O_2$ was preferentially broken down into $O_2$ and $H_2O$ when pH was higher than 3 (equation (1.7)) [27]. The decrease in the amount of $Fe^{2+}$ and $H_2O_2$ made less hydroxyl radicals generated (equation (1.5)), and then fewer organic matter in the ML-GFW were degraded.

The removal pathway of ML-GFW could be divided into two ways in this EF system, which were oxidation and flocculation. The oxidation was weakened along with pH value increasing from 3 to 8, but the flocculation was strengthened meanwhile. According to Ben's research [28], it was observed that at low pH values, iron is practically at the bivalent $Fe^{2+}$ or trivalent $Fe^{3+}$ states. When the pH increases, $Fe^{3+}$ and $Fe^{2+}$ react with $OH^-$ ions and the generated $[Fe(OH)_3]$ and $[Fe(OH)_2]$ during flocculation process remain in the aqueous stream as gelatinous suspension leading to the removal of organic pollutants. As a result, COD removal efficiency by flocculation increased when pH rose from 3 to 8, and the combination of oxidation and flocculation resulted in the CODR dropping slowly with pH value ranging from 3 to 8.

## 3.2. Effect of initial pH value on SEEC and SEPC

In the EF process, SEEC was one of the most important parameters for assessing the economic benefits in treating the ML-GFW. Effect of the initial pH on SEEC in the EF process was studied and the results are shown in figure 3a. After reaction for 3 h, the SEEC was 8.8, 4.7, 4.9, 5.8, 5.4, 5.6 and 7.3 kW h $kg_{COD}^{-1}$ when the initial pH varied from 2 to 8, respectively. Since there were few studies on the treatment of actual ML-GFW by EF process, the concentrated cutting oil diluent was chosen to be compared, and 24 kW h $kg_{COD}^{-1}$ was the lowest SEEC by EF process [16]. Two characteristics of the ML-GFW could account for the low SEEC in this study. One was the fairly high conductivity of the raw ML-GFW ($4.76 \times 10^5 \, \mu S \, cm^{-1}$). Since the SEEC decreased consequently and the corresponding COD removal raised rapidly when the conductivity increased [16], and the high conductivity of the raw ML-GFW had an indelible advantage in reducing energy consumption and material cost. The other was the fairly high value of COD concentration ($1.54 \times 10^4 \, mg \, l^{-1}$). The higher the pollutant concentration was,

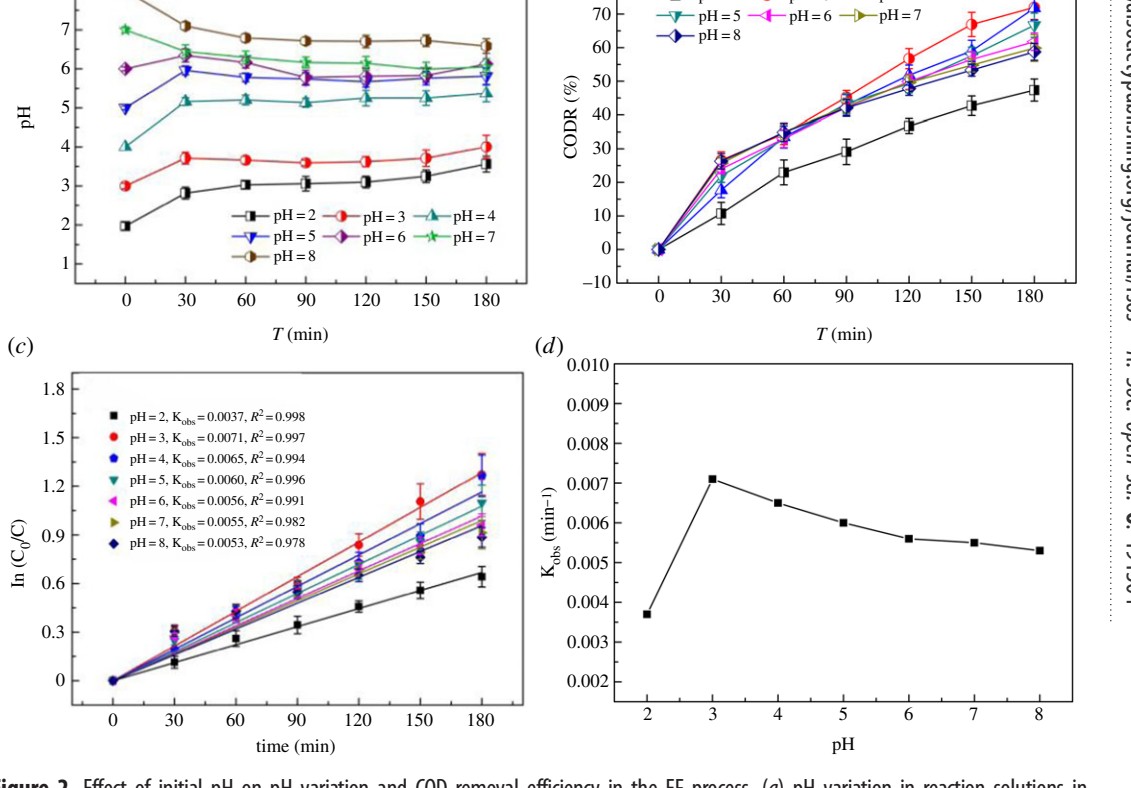

**Figure 2.** Effect of initial pH on pH variation and COD removal efficiency in the EF process. (a) pH variation in reaction solutions in EF process, (b) COD removal efficiency of ML-GFW, (c) COD removal efficiency described by the first-order equation, (d) $K_{obs}$ (min$^{-1}$) for ML-GFW removal efficiency. Conditions: 500 ml ML-GFW; I = 2 A; period reversal time: 10 min; H$_2$O$_2$: 7 ml/30 min; reaction time: 3 h.

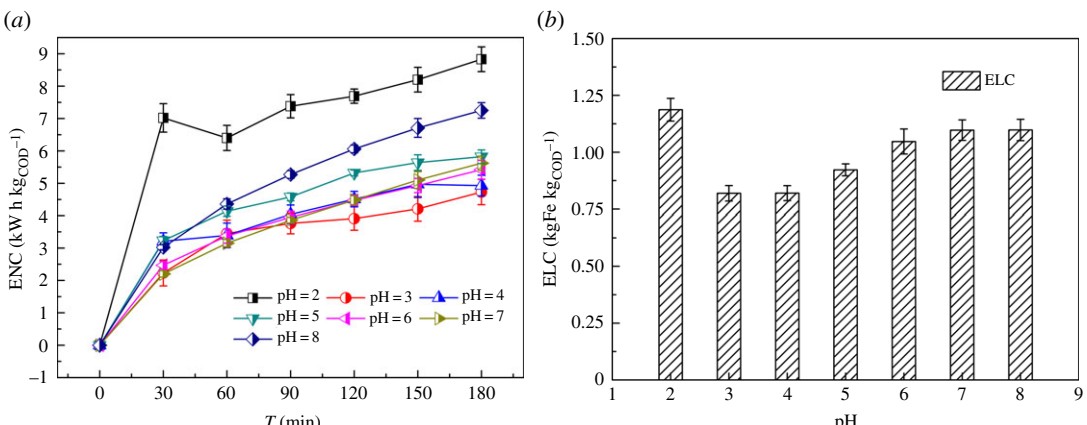

**Figure 3.** Effect of the initial pH on (a) SEEC in EF process, (b) SEPC in EF process. Conditions: 500 ml ML-GFW; I = 2 A; period reversal time: 10 min; H$_2$O$_2$: 7 ml/30 min; reaction time: 3 h.

the greater the amount of pollutants degraded would be, and the lower SEEC would be achieved. A constant current of 2.0 A was used to deal with the ML-GFW so that the power consumed in different initial pH values during the EF process did not change much, which was ranging from 0.02 to 0.027 kW h. Besides the effect of electrical power, SEEC has a direct relationship with COD removal performance. In figure 3a, the SEEC was the highest at pH of 2 since the COD removal efficiency through oxidation and flocculation was both low at the pH value. In all, the lowest SEEC was attained at pH of 3. Labiadh *et al*. [29] got a similar result treating concentrate from reverse osmosis of sanitary landfill leachate by an electro-chemical process.

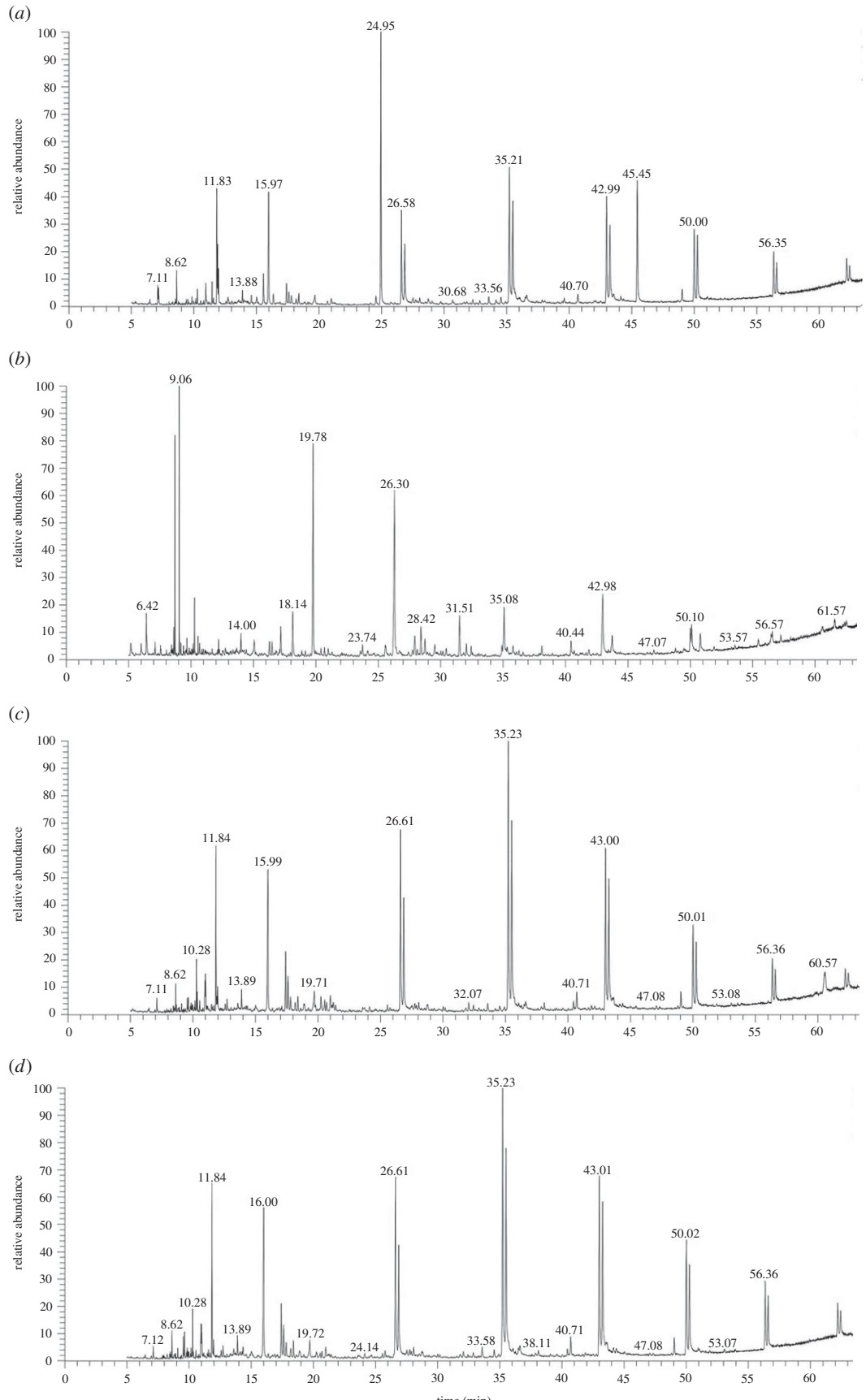

**Figure 4.** GC-MS analysis. (*a*) Raw ML-GFW, ML-GFW after treated at (*b*) pH of 3, (*c*) pH of 6, (*d*) pH of 8, and flocs eluent at (*e*) pH of 3, (*f*) pH of 6, (*g*) pH = 8 after 3 h reaction. Conditions: 500 ml ML-GFW; I = 2 A; period reversal time: 10 min; initial pH: 3, 6, 8; $H_2O_2$: 7 ml/30 min; reaction time: 3 h.

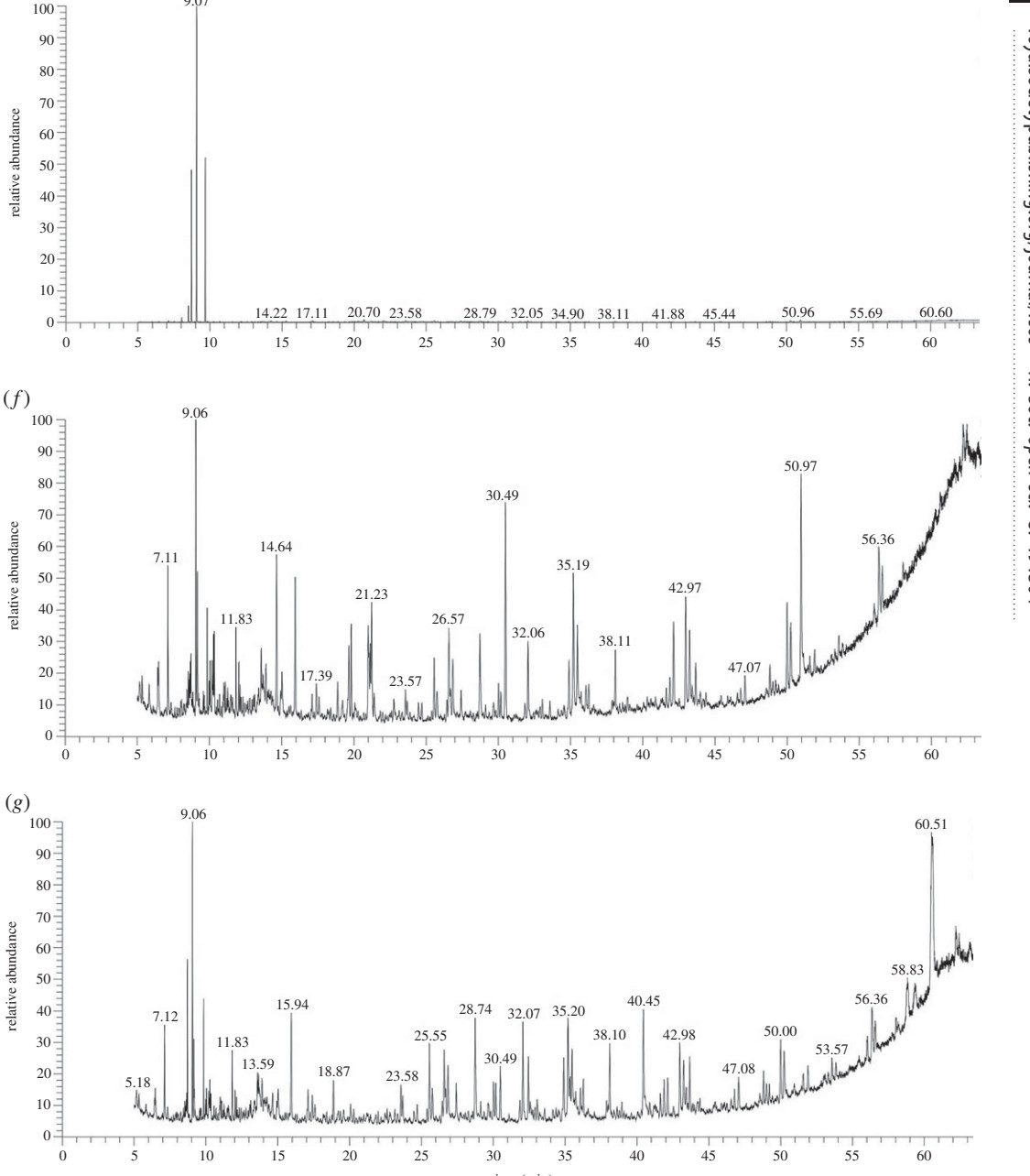

**Figure 4.** (Continued.)

SEPC was also an important index for assessing the economic benefits of the EF process in treating the ML-GFW. The results of SEPC under different pH values are shown in figure 3*b*. The SEPC values were 1.19, 0.82, 0.82, 0.92, 1.05, 1.10 and 1.10 $kgFe\ kg_{COD}^{-1}$ with initial pH value of 2, 3, 4, 5, 6, 7 and 8, respectively. The highest SEPC was achieved with pH of 2, which was about 1.3 times the lowest SEPC at pH of 3. Then the SEPC became higher as the initial pH increased from 4 to 8. According to equation (2.5), SEPC was the ratio of the quantity of dissolved electrode plate and the reducing COD concentration. Similar to the SEEC, the small change in the effect of dissolved electrode plate on the SEPC was limited because the variation of dissolved iron plates was small. With regard to SEEC and SEPC, pH value of 3 was the optimum choice considering economic benefits.

## 3.3. Oxidation and flocculation in the EF process

Organic pollutant removal from the ML-GFW can be owing to two main ways: oxidation and flocculation. On the one hand, the iron anode produced ferrous ions continuously (equation (1.1)),

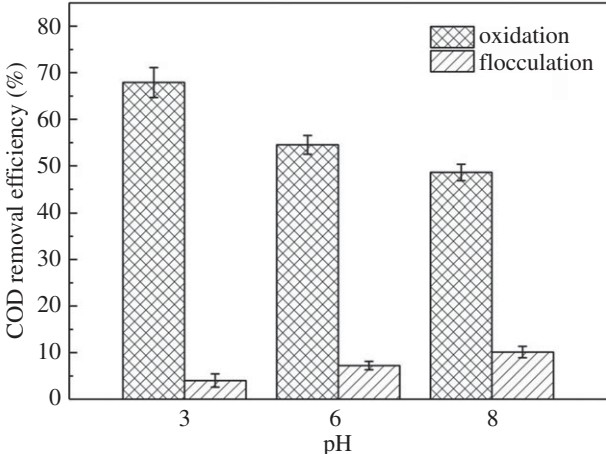

**Figure 5.** Effect of the initial pH on oxidation and flocculation in EF process. Conditions: 500 ml ML-GFW; I = 2A; period reversal time: 10 min; initial pH: 3, 6, 8; $H_2O_2$: 7 ml/30 min; reaction time: 3 h.

and 7 ml hydrogen peroxide was added every half an hour, thus there would be enough Fenton reagent ($H_2O_2/Fe^{2+}$) in the system. According to equation (1.5), a large number of hydroxyl radicals were continuously generated [30], which oxidized organic matter in the ML-GFW (equation (1.6)). On the other hand, a lot of ferrous ions generated on the iron anode after being energized (equation (1.1)). According to equations (1.8) and (1.9), a large number of flocs were produced with appropriate pH value, which could remove organic matter from ML-GFW by flocculation.

To study the performance of oxidation and flocculation during the EF process, pH of the ML-GFW was adjusted to the value of 3, 6 and 8. The samples taken from the raw ML-GFW, reaction solution and floc eluent were examined by GC-MS after 3 h reaction, and specific steps are shown in §2.3. Comparing the GC-MS analysis (figure 4a–d), the peak shapes of the treated ML-GFW at pH values of 6 and 8 were similar, which indicated that the products generated in the EF process were similar as well. More peaks were observed at pH of 3 compared with pH of 6 and 8, which could indicate that the products generated in the oxidation process were more diverse and the degradation mechanism seemed quite different from pH of 6 and 8. To take the GC-MS analyses of the floc eluent into consideration, figure 4e,f,g shows that few contaminants were adsorbed on the flocs at pH of 3. The results indicate that the acidic environment was not conducive to the flocculation during the EF process since the formation of iron hydroxide and ferrous hydroxide was weakened. When pH value was controlled at 6 and 8, the kind and quantity of contaminants adsorbed on the flocs at pH of 8 was more than pH of 6, which may be due to the difference in the quantity of flocs generated in different initial pH values. It can be speculated that the amount of flocs generated at pH of 8 was higher than pH of 6, since $Fe(OH)_3$ and $Fe(OH)_2$ represented the major fraction of the total dissolved iron at pH of 8 [31].

## 3.4. Mechanism in organic contaminant removal from the ML-GFW

COD in the raw ML-GFW, reaction solution and floc eluent were measured and calculated to represent organic contaminant removal efficiency through oxidation and flocculation, respectively. According to equations (2.1)–(2.3), COD removal efficiency of oxidation and flocculation could be calculated. Figure 5 shows the effect of initial pH value on the pathway of organic contaminant removal in the EF process. After reaction for 3 h, CODR was 67.9%, 54.5% and 48.6% through oxidation and was 4.0%, 7.2%, 10.1% through flocculation when pH values of the initial ML-GFW were 3.0, 6.0 and 8.0, respectively. The COD removal efficiency was similar to the research on dealing with leachate by EF with pH value of 3, and with the increase in solution pH, the COD removal through flocculation increased [32]. Therefore, oxidation was the major way to remove organic matter from the ML-GFW while flocculation had less effect on organic contaminant removal in the EF system.

Primary contaminants and non-volatile products generated in the EF system were identified by GC-MS in the samples. A typical chromatogram of the raw ML-GFW is presented in figure 4a, ML-GFW contained mainly various alkane, alcohol, amine, aromatic compounds and organic acids. The existence of different functional groups in the contaminants induced different degradation properties. The contaminants in the

(*a*)  totally oxidized when pH = 3, 6, 8

(*b*)  partially oxidized when pH = 3, 6, 8

(*c*)  oxidized only when pH = 3

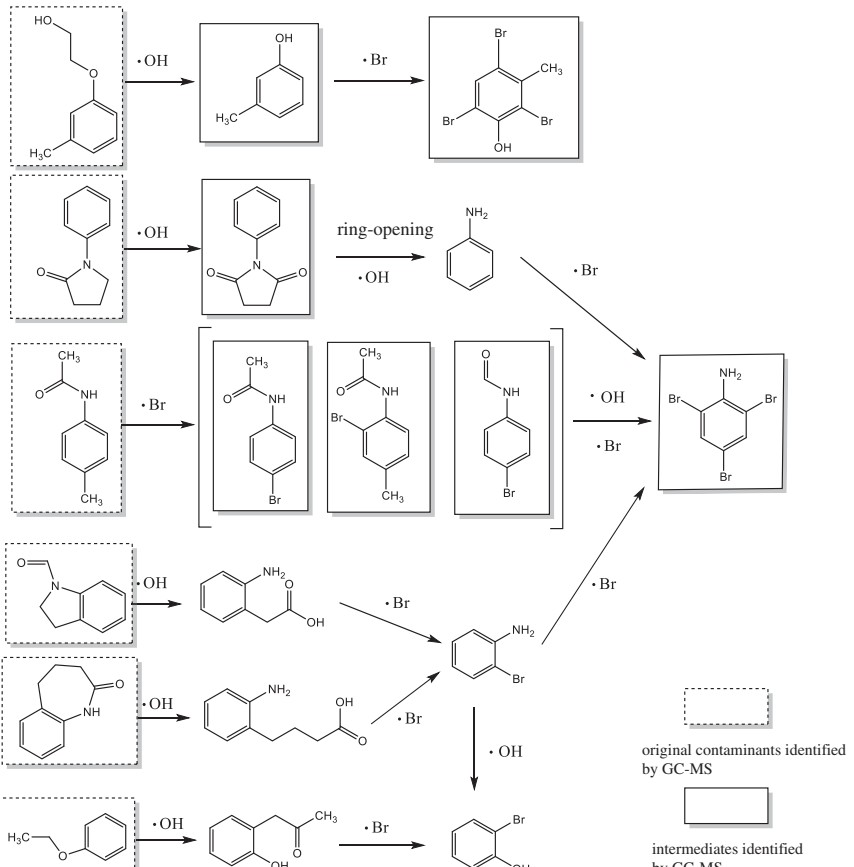

**Figure 6.** Proposed degradation pathways for major identified contaminants in ML-GFW in EF process.

ML-GFW were removed in two ways in the EF system: oxidation and flocculation. The following analysis of GC-MS results would be achieved from two aspects. For oxidation, the peak areas of contaminants in the reaction solution and floc eluent were superimposed at pH value of 3, 6 and 8, and compared with raw ML-GFW. In this way, the amount and kind of contaminants oxidized in the EF system could be clearly seen. The oxidation conversion pathways of some representative identified contaminants at different pH values are speculated in figure 6. As clarified in §3.1, the optimal pH value for oxidation is 3 and oxidation became weaken with the increase of pH value. It could be speculated that the amount and kind of contaminants oxidized were weakened with pH value rising from 3 to 8. The contaminants oxidized in the ML-GFW were classified into three parts. The first was totally oxidized at pH of 3, 6 and 8, and there could be only some simple heterocyclic organic compounds such as morpholine compounds including 4-morpholinecarbonylchloride (9.55 min) and N-(3-aminopropyl) morpholine (24.95 min), the proposed degradation pathways for the simple heterocyclic organic compounds are shown in figure 6a. The second part was the contaminants being partially oxidized at pH of 3, 6 and 8, which involved some heterocyclic organic matter without a benzene ring, and (1S,2R)-2-chloro-cyclopentanol (7.11 min), 3,3-diethyl-pyrrolidine-2,4-dione (8.62 min) and perhydrothiaxanthene (33.56 min) were included. What is more, the treatment rate of these contaminants at pH of 6 and 8 was much lower than at pH of 3. The proposed degradation pathways can be seen in figure 6b. The last part was the contaminants that can only be oxidized at pH of 3, which were some aromatic compounds with some specific functional groups. Several representative contaminants involved 2-(3-methylphenoxy)ethanol (11 min), 4-methylacetanilide (13.88 min), 1H-indole-1-carboxaldehyde (15.97 min), 1,3,4,5-tetrahydro-benzo[b]azepin-2-one (17.79 min), 1-phenyl-2-pyrrolidinone (18.39 min) and 2,3-dihydro-2-methyl-benzofuran (26.58 min). The proposed degradation pathways are shown in figure 6c. As for flocculation, according to the analysis for figure 4a,e,f,g, long-chain alkane compounds, such as palmitic acid and stearic acid, were the main organic contaminants removed by flocculation.

## 4. Conclusion

The EF process was applied in treating the ML-GFW. Different initial pH values had great influence on CODR, SEEC, SEPC and degradation pathways. COD removal efficiency of 71.9% was achieved at pH of 3, current of 2.0 A, 7 ml of $H_2O_2$ every half an hour and reaction time of 3 h. Organic contaminants in the ML-GFW were removed through two pathways: oxidation and flocculation. Compared with flocculation, oxidation was the major pathway to remove organic matter in the EF process, which was enhanced at pH of 3. SEEC and SEPC were the lowest at pH of 3 and the values were 4.73 kW h $kg_{COD}^{-1}$ and 0.82 kgFe $kg_{COD}^{-1}$. Therefore, pH of 3 was the optimum choice in treating the ML-GFW considering organic contaminant removal performance and the cost. Through the analysis of GC-MS, long-chain alkane compounds were more likely to be removed by flocculation and the degradation pathways of contaminants in ML-GFW were various in different initial pH values. Some complex contaminants can only be oxidized at pH of 3, such as aromatic compounds with some specific functional groups. EF process can be used successfully in the pretreatment of ML-GFW.

Data accessibility. The data have been submitted as electronic supplementary material.
Authors' contributions. Y.W. mainly carried out experiments, analysed data and wrote the manuscript. H.-q.L. is the corresponding author and he mainly guided experiments, analysed data and revised the article. Li-ming Ren assisted the experiments and analyse data.
Competing interests. We declare we have no competing interests.
Funding. This research was supported by the International Scientific and Technological Innovation and Cooperation Project of Sichuan (grant no. 2019YFH0170).

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
