## [Reviewer comments · Royal Society Open Science]

Review History

RSOS-191304.R0 (Original submission)

Review form: Reviewer 1

Is the manuscript scientifically sound in its present form?

No

Are the interpretations and conclusions justified by the results?

No

Is the language acceptable?

No

Do you have any ethical concerns with this paper?

No

Have you any concerns about statistical analyses in this paper?

No

Recommendation?

Reject

Comments to the Author(s)

The authors studied the effect of initial pH on the electro-Fenton treatment of ML-GFW. They concluded the pH of 3 was the optimum choice. In addition, the organic matter removal mechanism was proposed. However, the conclusions of this paper are not justified by the results. The differences in the EF performance between pH of 3 and pH of 4 are negligible, which are within the error ranges. The authors only showed the error bars in Fig. 2 (a) and (b) and did not plot the error bar when calculating the Kobs. Furthermore, the fitting curve in Fig. 2 (c) is apparently not correct because the red curve is above all the data points. I think this paper is not appropriate for publication in RSOS.

Review form: Reviewer 2**Is the manuscript scientifically sound in its present form?**

Yes

Are the interpretations and conclusions justified by the results?

Yes

Is the language acceptable?

No

Do you have any ethical concerns with this paper?

No

Have you any concerns about statistical analyses in this paper?

No

Recommendation?

Accept with minor revision (please list in comments)

Comments to the Author(s)

The authors demonstrated the electro-Fenton process for treatment of mother liquor of gas field waste water (ML-GFW). Organic matter removal was achieved in two ways: oxidation and flocculation, and oxidation played a major role in this study. The manuscript is well written, and the quality of article is good in my opinion. Manuscript can be accepted after minor revision.

- 1) English grammar, spacing should be checked out throughout the manuscript.
- 2) How is the manuscript advancing the present field? What is new in the manuscript compare to previous studies? Authors can highlight these points in the introduction part of the manuscript.
- 3) Page 6, equation 7 should rewrite correctly. Use O to represent oxygen in water instead of zero 0.
- 4) Figure 4 is unreadable. Authors can rearrange the figure in order to increase visibility.
- 5) In the case of flocculation, COD removal efficiency increases on increasing pH, why? Give more explanation.

Review form: Reviewer 3

Is the manuscript scientifically sound in its present form?

Yes

Are the interpretations and conclusions justified by the results?

Yes

Is the language acceptable?

Yes

Do you have any ethical concerns with this paper?

No

Have you any concerns about statistical analyses in this paper?

No

Recommendation?

Accept with minor revision (please list in comments)

Comments to the Author(s)

The authors reported an electron-Fenton process for organic wastewater treatment. The idea is interesting and the performance is good. This work may provide a new efficient way to treat organic wastewater. After Minor revision, it is suitable to be published.

1. Several typing errors should be corrected.
2. The electrochemical process as well as the electrochemical CV curves should provide.
3. The evidence for the existence of .hydroxide radicals should be given.

Decision letter (RSOS-191304.R0)

23-Sep-2019

Dear Dr Li:

Title: Organic matters removal from mother liquor of gas field wastewater by electro-Fenton process with the addition of H₂O₂: Effect of initial pH

Manuscript ID: RSOS-191304

The editor assigned to your manuscript has now received comments from reviewers. We would like you to revise your paper in accordance with the referee and Subject Editor suggestions which can be found below (not including confidential reports to the Editor). Please note this decision does not guarantee eventual acceptance.

Please submit your revised paper before 16-Oct-2019. Please note that the revision deadline will expire at 00.00am on this date. If we do not hear from you within this time then it will be assumed that the paper has been withdrawn. In exceptional circumstances, extensions may be possible if agreed with the Editorial Office in advance. We do not allow multiple rounds of

revision so we urge you to make every effort to fully address all of the comments at this stage. If deemed necessary by the Editors, your manuscript will be sent back to one or more of the original reviewers for assessment. If the original reviewers are not available we may invite new reviewers.

Please also include the following statements alongside the other end statements. As we cannot publish your manuscript without these end statements included, if you feel that a given heading is not relevant to your paper, please nevertheless include the heading and explicitly state that it is not relevant to your work.

- Acknowledgements

- Funding statement

Please include a funding section after your main text which lists the source of funding for each author.

RSC Associate Editor:
Comments to the Author:
(There are no comments.)

RSC Subject Editor:
Comments to the Author:
(There are no comments.)

Reviewers' Comments to Author:
Reviewer: 1

Comments to the Author(s)

The authors studied the effect of initial pH on the electro-Fenton treatment of ML-GFW. They concluded the pH of 3 was the optimum choice. In addition, the organic matter removal mechanism was proposed. However, the conclusions of this paper are not justified by the results. The differences in the EF performance between pH of 3 and pH of 4 are negligible, which are within the error ranges. The authors only showed the error bars in Fig. 2 (a) and (b) and did not plot the error bar when calculating the Kobs. Furthermore, the fitting curve in Fig. 2 (c) is apparently not correct because the red curve is above all the data points. I think this paper is not appropriate for publication in RSOS.

Reviewer: 2

Comments to the Author(s)

The authors demonstrated the electro-Fenton process for treatment of mother liquor of gas field waste water (ML-GFW). Organic matter removal was achieved in two ways: oxidation and flocculation, and oxidation played a major role in this study. The manuscript is well written, and the quality of article is good in my opinion. Manuscript can be accepted after minor revision.

- 1) English grammar, spacing should be checked out throughout the manuscript.
- 2) How is the manuscript advancing the present field? What is new in the manuscript compare to previous studies? Authors can highlight these points in the introduction part of the manuscript.
- 3) Page 6, equation 7 should rewrite correctly. Use O to represent oxygen in water instead of zero 0.
- 4) Figure 4 is unreadable. Authors can rearrange the figure in order to increase visibility.
- 5) In the case of flocculation, COD removal efficiency increases on increasing pH, why? Give more explanation.

Reviewer: 3

Comments to the Author(s)

The authors reported an electron-Fenton process for organic wastewater treatment. The idea is interesting and the performance is good. This work may provide a new efficient way to treat organic wastewater. After Minor revision, it is suitable to be published.

1. Several typing errors should be corrected.
2. The electrochemical process as well as the electrochemical CV curves should provide.
3. The evidence for the existence of .hydroxide radicals should be given.

Author's Response to Decision Letter for (RSOS-191304.R0)

See Appendix A.

RSOS-191304.R1 (Revision)

Review form: Reviewer 2

Is the manuscript scientifically sound in its present form?

Yes

Are the interpretations and conclusions justified by the results?

Yes

Is the language acceptable?

Yes

Do you have any ethical concerns with this paper?

No

Have you any concerns about statistical analyses in this paper?

No

Recommendation?

Accept as is

Comments to the Author(s)

With considering the interdisciplinary readers, authors should add the full form of COD when they described it the first time.

Decision letter (RSOS-191304.R1)

28-Oct-2019

Dear Dr Li:

Title: Organic matters removal from mother liquor of gas field wastewater by electro-Fenton process with the addition of H₂O₂: Effect of initial pH

Manuscript ID: RSOS-191304.R1

It is a pleasure to accept your manuscript in its current form for publication in Royal Society Open Science. The chemistry content of Royal Society Open Science is published in collaboration with the Royal Society of Chemistry.

Yours sincerely,
Dr Laura Smith

Publishing Editor, Journals

RSC Associate Editor:
Comments to the Author:
Please include the full form of the abbreviation "COD" before final publication.

RSC Subject Editor:
Comments to the Author:
(There are no comments.)

Reviewer(s)' Comments to Author:
Reviewer: 2

Comments to the Author(s)
With considering the interdisciplinary readers, authors should add the full form of COD when they described it the first time.

Appendix A

Dear Editor

Thank you very much for your letter and the comments from the referees about our paper submitted to Royal Society Open Science (Manuscript ID: RSOS-191304). Hope these will make it more acceptable for publication.

Response to Reviewer 1:

- The authors studied the effect of initial pH on the electro-Fenton treatment of ML-GFW. They concluded the pH of 3 was the optimum choice. In addition, the organic matter removal mechanism was proposed. However, the conclusions of this paper are not justified by the results. The differences in the EF performance between pH of 3 and pH of 4 are negligible, which are within the error ranges. The authors only showed the error bars in Fig. 2 (a) and (b) and did not plot the error bar when calculating the K_{obs} .

Response: Fig.2(d) has been redrawn, and the k_{obs} of pH=2, 3, 4, 5, 6, 7, 8 are 0.0037, 0.0071, 0.0065, 0.0060, 0.0056, 0.0055, 0.0053. The results of k_{obs} indicated that the EF performance at pH of 3 was better than at pH of 4.

- Furthermore, the fitting curve in Fig. 2 (c) is apparently not correct because the red curve is above all the data points.

Response: Fig. 2 (c) has been redrawn according to the uploaded data from the supplementary material and the error bars were added.

Response to Reviewer 2:

- English grammar, spacing should be checked out throughout the manuscript.

Response: English grammar, spacing have been checked out throughout the manuscript.

- How is the manuscript advancing the present field? What is new in the manuscript compare to previous studies? Authors can highlight these points in the introduction part of the manuscript.

The statement of “Compare to previous studies, there was less related research on treating ML-GFW by electro-Fenton. The research provided a theoretical basis for the application of EF process in treating actual ML-GFW.” has been added on page 5, lines 8-11.

- Page 6, equation 7 should rewrite correctly. Use O to represent oxygen in water instead of zero 0.

Response: The chemical formula for water (H_2O) in page 6, equation 7 had been corrected.

- Figure 4 is unreadable. Authors can rearrange the figure in order to increase visibility.

Response: Fig. 4 had been arranged in a column for visibility.

- In the case of flocculation, COD removal efficiency increases on increasing pH, why? Give more explanation.

Response: The statement of “According to Ben Hariz’s research, it was observed that low pH values...” has been added on page 10, lines 12-17.

Response to Reviewer3:

- Several typing errors should be corrected.

Response: We have checked the manuscript carefully, and typing errors have been corrected.

- The electrochemical process as well as the electrochemical CV curves should provide.

Response: The statement of “Iron electrodes were one of the most common materials...” is shown on page 5, lines 17-19.

- The evidence for the existence of hydroxide radicals should be given.

Response: Iron electrodes were used in the experiments, which could produce sufficient ferrous ions during the electrolysis, and the H_2O_2 was added into the EF

system directly. According to the following equation, the hydroxide radicals must exist in the system.